# StreamFlow: Theory, Algorithm, and Implementation for High-Efficiency Rectified Flow Generation

**Sen Fang** [1] [*]   **Hongbin Zhong** [2] [*]   **Yalin Feng** [3]   **Yanxin Zhang** [4]   **Dimitris N. Metaxas** [1]

## Abstract

New technologies such as Rectified Flow and Flow Matching have significantly improved the performance of generative models in the past two years, especially in terms of control accuracy, generation quality, and generation efficiency. However, due to some differences in its theory, design, and existing diffusion models, the existing acceleration methods cannot be directly applied to the Rectified Flow model. In this article, we have comprehensively implemented an overall acceleration pipeline from the aspects of theory, design, and reasoning strategies. This pipeline uses new methods such as batch processing with a new velocity field, vectorization of heterogeneous time-step batch processing, and dynamic TensorRT compilation for the new methods to comprehensively accelerate related models based on flow models. Currently, the existing public methods usually achieve an acceleration of 18%, while experiments have proved that our new method can accelerate the $512\times512$ image generation speed to up to 611%, which is far beyond the current non-generalized acceleration methods. Project page at https://world-snapshot.github.io/StreamFlow/.

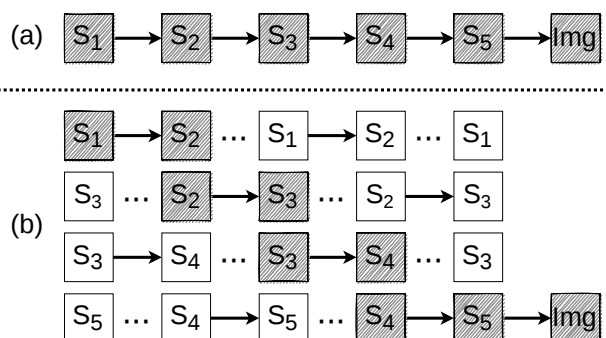

*Figure 1.* **The differences between original denoising and batch denoising:** We can consider each line as a separate parallel operation queue. **(a)** This is the original diffusion generation process. Suppose we need 5 steps to complete it, then denoising is sequential. **(b)** Then some common acceleration methods involve using multiple parallel operation queues, with each queue only considering the requirements of a certain step. The grey area represents the complete process of a certain generation.

## 1. Introduction

**Rectified Flow** (Liu et al., 2022) and **Flow Matching** (Lipman et al., 2022) are new generation methods that became popular about two years ago. Existing generation processes (such as *diffusion* (Ho et al., 2020; Song et al., 2020)) generally learn to gradually shift from a Gaussian noise distribution to the real distribution/image distribution. They usually require *several dozen steps* to predict the next noise from the previous noise. However, these two new methods attempt to provide a *velocity field*, allowing the model to directly predict the direction of the next distribution and the speed for each step, and then try to learn a more direct path between the distributions, enabling good results to be achieved in **4-step generation** (Yan et al., 2024; Liu et al., 2024).

Previously, the generation based on diffusion models has had many limitations in terms of theory and mechanism, resulting in relatively slow generation efficiency. With the popularity of diffusion models in areas such as augmented/virtual reality, style transfer, and real-time action generation, there have been works called **StreamDiffusion** (Kodaira et al., 2025) that attempt to solve this problem. As shown in Fig. 1, they achieve this by creating a *pipeline*, by decoupling noise addition and denoising stages and mapping them to parallel pipeline queues, which makes it particularly fast in accelerating the generation of diffusion models.

However, when attempting to transfer these diffusion-oriented acceleration frameworks to Rectified Flow models, we encounter fundamental mismatches at the level of representation, scheduling semantics, and execution structure:

---

[*]Equal contribution [1]Rutgers University, New Jersey, USA [2]Georgia Institute of Technology, Atlanta, Georgia, USA [3]Nanyang Technological University, Singapore [4]University of Wisconsin-Madison, Wisconsin, USA. Correspondence to: Sen Fang <sen.fang@rutgers.edu>.

*Proceedings of the 43rd International Conference on Machine Learning*, Seoul, South Korea. PMLR 306, 2026. Copyright 2026 by the author(s).

**(1)** As mentioned in the first paragraph, their theories are not consistent. The content they generate at each step and the process of generating also differ. For example, they deal with *time fields* rather than noise. **(2)** In terms of scheduling strategies and other policies, they use the *time step* to dynamically divide the progress of the content and then guide the generation process. **(3)** When we attempt to make them compatible with the existing compilation-based acceleration methods (NVIDIA Corporation, 2024), due to their distinct structures and the fact that they change the structure depending on the settings, this often leads to *compilation failures*.

To address these issues, we first, based on the generation theory of the Flow model, attempted to create a *batch processing of velocity fields*. For different time steps as inputs, they were divided into different processing steps, conceptually treating them as a type of **operation step**. Secondly, we developed *vectorized heterogeneous time-step batch processing* for these operation steps. According to the sequence of these different times, they were dynamically filled into the pipeline for processing. Finally, we regarded the entire framework as a dynamic and structurally changing generation process, and further accelerated it using TensorRT (NVIDIA Corporation, 2024) compilation.

However, when we actually implemented it, some new challenges emerged[1]:**(1)** Rectified Flow employs variable-length time steps, leading to misaligned and non-uniform processing stages. **(2)** The existing scheduler wrapper is unable to handle the *vectorized output* and *differentiable time steps*. **(3)** Even if TensorRT cannot simply achieve the functions we envisioned, the compilation is not only slow but also prone to crashing.

Therefore, we developed a batch processing system for the velocity field and made it compatible. Based on the theory of the flow model, we regarded the velocity field as *random step-length noise steps*. Then we **decoupled the binding** of the noise steps with the pipeline, allowing the parallel queues to have any step-length velocities, and developed a scheduler that can handle such situations. For compilation-based acceleration, we added model construction compilation to automatically process structural information during compilation, and then our compilation script *adaptively processes* this information for compilation.

Therefore, our contributions can be summarized as follows:

- **Large-scale batch velocity field processing** with operational steps that *decouple the binding* between step size and velocity field, featuring an adaptive pipeline with queue management for continuous velocity field output.

---

[1]These new challenges are specific to flow models and cannot be solved by any traditional acceleration framework.

- A **scheduler that vectorizes heterogeneous timestep batches**, enabling unified processing of outputs with varying step sizes for collaborative velocity field generation and flow model architectures.

- **Dynamic singular structure TensorRT compilation** with *plug-and-play compatibility* for flow model characteristics, further accelerating flow models within our acceleration framework.

## 2. Related Works

**Rectified Flow and Flow Matching:** Flow Matching (FM) and Rectified Flow are recent paradigms that bridge diffusion models and continuous normalizing flows by directly regressing a velocity field $\mathbf{v}_\theta(\mathbf{x}_t, t)$ that maps Gaussian noise $\mathbf{x}_0 \sim \mathcal{N}(0, I)$ to data $\mathbf{x}_1 \sim p_{\text{data}}$ via the ODE $d\mathbf{x}_t = \mathbf{v}_\theta(\mathbf{x}_t, t)dt$ (Lipman et al., 2022; Albergo & Vanden-Eijnden, 2022). The key innovation lies in defining conditional probability paths that minimize transport cost. FM enables the use of optimal transport (straight-line) paths $\mathbf{x}_t = (1 - t)\mathbf{x}_0 + t\mathbf{x}_1$, trained by minimizing the regression objective $\mathcal{L} = \mathbb{E}_{t,\mathbf{x}_0,\mathbf{x}_1} \|\mathbf{v}_\theta(\mathbf{x}_t, t) - (\mathbf{x}_1 - \mathbf{x}_0)\|^2$ (Lipman et al., 2022). This formulation allows for fast inference through simple Euler integration $\mathbf{x}_1 \approx \mathbf{x}_0 + \mathbf{v}_\theta(\mathbf{x}_0, 0)$ in as few as one to four steps. Building on this, methods like InstaFlow successfully apply text-conditioned reflow to large-scale Stable Diffusion models, achieving high-quality one-step generation (Liu et al., 2024). These advancements lay the theoretical groundwork for our accelerated pipeline.

**Generative Image Models:** Recent progress in generative modeling has been driven by diffusion, transformer, and flow-based architectures. DDPM (Ho et al., 2020) set the foundation for high-fidelity synthesis through iterative denoising, albeit with slow sampling. Latent Diffusion Models (LDM) (Rombach et al., 2022a) reduce computation by operating in compressed latent space, while Diffusion Transformers (DiT) (Peebles & Xie, 2023) replace U-Nets with Vision Transformers to enhance scalability. Beyond classical diffusion, hybrid designs such as VA-VAE (Yao et al., 2025) and RAC (Fang et al., 2026b) employ pretrained vision models to regularize high-dimensional latents, alleviating the reconstruction–generation trade-off. For fast sampling, PeRFlow (Yan et al., 2024) accelerates inference via piecewise linear flow segments, functioning as a plug-and-play module. Additionally, Visual Autoregressive Modeling (VAR) (Tian et al., 2024) emerges as a scalable alternative with a next-scale prediction scheme. Some work (Chen et al., 2024; Fang et al., 2025b;c; 2026c) has also adopted the ideas of Flow to varying degrees. These advances motivate our new rectified flow acceleration framework.

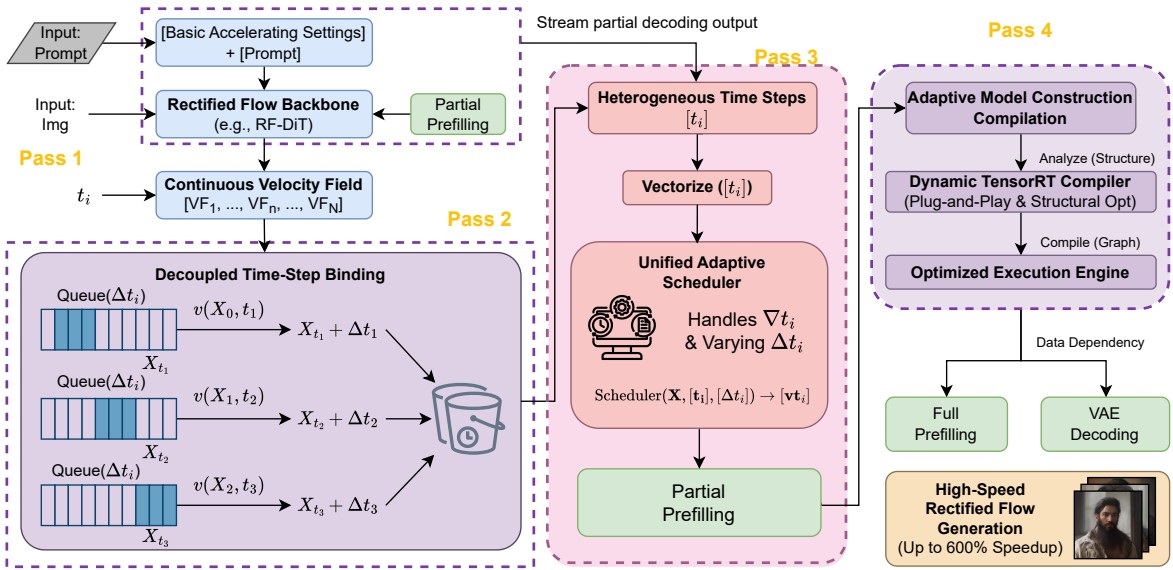

*Figure 2.* **Overview of our StreamFlow pipeline: (Pass 1)** The process initiates with prompt processing via the Rectified Flow Backbone and partial prefilling to establish the continuous velocity field $v_t(X_t)$ based on the trajectory equation $dX_t = v_t(X_t)dt$. **(Pass 2)** The Decoupled Time-Step Binding mechanism manages parallel queues with heterogeneous step sizes (e.g., $\Delta t_a$, $\Delta t_b$), independently applying velocity updates before converging via in-place aggregation for arbitrary batching. **(Pass 3)** Heterogeneous time steps $[t_i]$ are vectorized and managed by the Unified Adaptive Scheduler, which handles varying temporal gradients and step sizes to synchronize velocity outputs for subsequent partial prefilling. **(Pass 4)** We undergoes Adaptive Model Construction Compilation, utilizing a Dynamic TensorRT Compiler for plug-and-play compatibility and structural optimization to create an Optimized Execution Engine.

**Inference Acceleration:** To enable interactive generation (Fang et al., 2025a; 2026a), researchers have explored inference acceleration via step reduction optimizations. Multi-step diffusion models can be distilled to few-step or one-step generation (Liu et al., 2022; 2024), while ODE-based solvers (Liu et al., 2022; Wu et al., 2025) reframe diffusion as neural ODEs for faster integration. Engineering optimizations like model quantization and TensorRT (NVIDIA Corporation, 2024) can fuse layers and roughly double throughput by optimizing execution graphs. However, these assume fixed computation graphs and break under *dynamic time steps* or changing batch sizes common in Flow methods.

We extend these ideas to rectified flow by designing fully vectorized, batched scheduling of continuous time steps and integrating dynamic TensorRT compilation to handle variable step sizes and scheduler incompatibilities that diffusion-specific accelerators do not face. Our unified pipeline boosts rectified flow generation by up to 611%, far surpassing prior general-purpose methods.

## 3. Methodology

### 3.1. Preliminaries and Problem Statement

#### 3.1.1. RECTIFIED FLOW AND VELOCITY FIELDS

Rectified Flow (Liu et al., 2022; Esser et al., 2024) formulates the generative process as learning a time-dependent

velocity field $v_\theta(x_t, t)$ that transports samples from a noise distribution $\pi_0$ to the data distribution $\pi_1$ along straight paths. The flow is defined by the ODE $\frac{dx_t}{dt} = v_\theta(x_t, t)$ for $t \in [0, 1]$. The model learns to predict the velocity field by minimizing:

$$\mathcal{L} = \mathbb{E}_{x_0, x_1, t} \left[ \|v_\theta(x_t, t) - (x_1 - x_0)\|^2 \right] \quad (1)$$

where $x_t = tx_1 + (1 - t)x_0$ represents the linear interpolation path. During inference, we discretize the continuous flow using $N$ steps with timesteps $\{t_i\}_{i=0}^{N-1}$: $x_{t_{i+1}} = x_{t_i} + \Delta t_i \cdot v_\theta(x_{t_i}, t_i)$ where $\Delta t_i = t_{i+1} - t_i$ is the step size.

#### 3.1.2. TIME-WINDOWS MECHANISM IN PERFLOW

PeRFlow (Yan et al., 2024) introduces a time-windows mechanism that divides the time interval $[0, 1]$ into $K$ non-overlapping windows $\mathcal{W}_k = [t_k^s, t_k^e]$. For a given timestep $t_c \in \mathcal{W}_k$, the velocity is computed using window-specific parameters:

$$\gamma_{s \to e} = \left( \frac{\alpha_{t_k^s}}{\alpha_{t_k^e}} \right)^{1/2} \quad (2)$$

$$\lambda_t = \frac{\lambda_s (t_k^e - t_k^s)}{\lambda_s (t_c - t_k^s) + (t_k^e - t_c)} \quad (3)$$

$$\eta_t = \frac{\eta_s(t_k^e - t_c)}{\lambda_s(t_c - t_k^s) + (t_k^e - t_c)} \quad (4)$$

where $\alpha_t$ represents the cumulative product of noise schedule coefficients, and $\lambda_s, \eta_s$ are window-dependent scaling factors derived from $\gamma_{s \to e}$. The predicted velocity is then $v_\theta(x_t, t) = (x_{pred}^{t_k^e} - x_t)/(t_k^e - t_c)$.

### 3.1.3. CHALLENGES IN BATCH ACCELERATION

Prior work on accelerating diffusion models (Lyu et al., 2022), such as StreamDiffusion (Kodaira et al., 2025), relies on batch denoising where all samples in a batch share the *same timestep* $t$. This homogeneous batching allows for $\mathbf{X}_{t'} = f_\theta(\mathbf{X}_t, t)$ where $\mathbf{X}_t \in \mathbb{R}^{B \times C \times H \times W}$. However, this approach fails for Rectified Flow models due to three fundamental challenges:

**Challenge 1: Velocity Field Batching.** Unlike noise prediction in diffusion models, velocity fields have varying step sizes $\Delta t_i$ that depend on the time-window structure. Standard schedulers cannot handle batched computations with different $\Delta t_i$ values.

**Challenge 2: Heterogeneous Timesteps.** Pipeline-based acceleration requires processing samples at different denoising stages simultaneously, resulting in heterogeneous timestep batches $\mathbf{t} = [t_0, t_1, \ldots, t_{N-1}]$ where $t_i \neq t_j$ for $i \neq j$.

**Challenge 3: Dynamic Compilation.** TensorRT and similar compilers assume static computational graphs. The time-window mechanism in Eq. 4 creates dynamic structures that violate this assumption, causing compilation failures.

### 3.2. StreamFlow: Architecture Overview

We propose StreamFlow, a comprehensive acceleration framework that addresses these challenges through three key innovations: (1) batched velocity field computation with vectorized time-windows, (2) heterogeneous timestep pipeline batching, and (3) dynamic TensorRT compilation. Figure 2 illustrates the overall pipeline.

As shown in Fig. 2, the key insight is to *decouple* the velocity field computation from step size constraints, enabling vectorized processing of heterogeneous timesteps while maintaining algorithmic correctness. This allows us to construct a pipeline where each UNet invocation processes $N$ samples at different denoising stages:

$$[\mathbf{x}_{t_0}, \mathbf{x}_{t_1}, \ldots, \mathbf{x}_{t_{N-1}}] \xrightarrow{UNet} [\mathbf{x}'_{t_0}, \mathbf{x}'_{t_1}, \ldots, \mathbf{x}'_{t_{N-1}}] \quad (5)$$

### 3.3. Batched Velocity Field Computation

#### 3.3.1. DECOUPLING STEP SIZE FROM VELOCITY FIELD

The vanilla PeRFlow (Yan et al., 2024) scheduler processes timesteps sequentially, computing window parameters for each $t_i$ individually. We reformulate this as a vectorized operation that processes a batch of timesteps $\mathbf{t} = [t_0, t_1, \ldots, t_{B-1}]$ simultaneously.

First, we vectorize the window lookup operation. For each timestep $t_i$, we identify its corresponding window $\mathcal{W}_{k_i}$ through $k_i = \arg\max_k\{t_i > t_k^e + \epsilon\}$ where $\epsilon$ is a numerical precision tolerance. This can be vectorized using masking operations: $\mathbf{M}_k = (\mathbf{t} > t_k^e + \epsilon)$ and $\mathbf{k} = \sum_{j=1}^{K} \mathbf{M}_j$.

#### 3.3.2. VECTORIZED WINDOW PARAMETER CALCULATION

Given the window assignments $\mathbf{k}$, we compute all window parameters in parallel. For a batch of $B$ timesteps, we construct the window boundaries $\mathbf{t}^s = [t_{k_0}^s, \ldots, t_{k_{B-1}}^s]$ and $\mathbf{t}^e = [t_{k_0}^e, \ldots, t_{k_{B-1}}^e]$, then compute $\boldsymbol{\gamma} = (\boldsymbol{\alpha}(\mathbf{t}^s)/\boldsymbol{\alpha}(\mathbf{t}^e))^{1/2}$. The batch velocity field parameters are computed as:

$$\boldsymbol{\lambda}_s = \boldsymbol{\gamma}^{-1}, \quad \boldsymbol{\eta}_s = -\frac{(1 - \boldsymbol{\gamma}^2)^{1/2}}{\boldsymbol{\gamma}},$$
$$\text{denom} = \boldsymbol{\lambda}_s \odot (\mathbf{t} - \mathbf{t}^s) + (\mathbf{t}^e - \mathbf{t}),$$
$$\boldsymbol{\lambda}_t = \frac{\boldsymbol{\lambda}_s \odot (\mathbf{t}^e - \mathbf{t}^s)}{\text{denom}}, \quad \boldsymbol{\eta}_t = \frac{\boldsymbol{\eta}_s \odot (\mathbf{t}^e - \mathbf{t})}{\text{denom}} \quad (6)$$

where $\odot$ denotes element-wise multiplication.

#### 3.3.3. BATCH SCHEDULER DESIGN

We extend the standard scheduler with a `step_batch` method that processes heterogeneous timesteps. For model outputs $\boldsymbol{\epsilon}_\theta \in \mathbb{R}^{B \times C \times H \times W}$ and latents $\mathbf{X} \in \mathbb{R}^{B \times C \times H \times W}$, we compute the predicted window endpoints $\mathbf{X}_{pred}^e = \boldsymbol{\lambda}_t \odot \mathbf{X} + \boldsymbol{\eta}_t \odot \boldsymbol{\epsilon}_\theta$, the velocity field $\mathbf{v}_\theta = (\mathbf{X}_{pred}^e - \mathbf{X})/(\mathbf{t}^e - \mathbf{t})$, and the next latents $\mathbf{X}_{next} = \mathbf{X} + \Delta\mathbf{t} \odot \mathbf{v}_\theta$ where $\Delta\mathbf{t} = \mathbf{t}_{next} - \mathbf{t}$.

Algorithm 1 presents the complete batched velocity field computation.

**Complexity Analysis.** The vanilla scheduler has time complexity $O(N \cdot T)$ for $N$ images with $T$ denoising steps each. Our batched scheduler reduces this to $O((N + T) \cdot K)$ where $K$ is the number of time windows (typically $K = 4$), achieving significant speedup when $N \gg K$.

### 3.4. Heterogeneous Timestep Pipeline Batching

#### 3.4.1. PIPELINE ARCHITECTURE AND BUFFER MANAGEMENT

We construct a pipeline that processes $N$ samples at different denoising stages concurrently. Let $S_i$ denote the

---

**Algorithm 1** Batched Velocity Field Step

---

**Require:** Model output $\boldsymbol{\epsilon}_\theta$, latents $\mathbf{X}$, timesteps $\mathbf{t}$
**Ensure:** Updated latents $\mathbf{X}_{next}$
1: $\mathbf{t}_c \leftarrow \mathbf{t}/T_{max}$ {Normalize to $[0, 1]$}
2: Compute window assignments: $\mathbf{t}^s, \mathbf{t}^e \leftarrow$ WindowLookup($\mathbf{t}_c$)
3: $\boldsymbol{\gamma} \leftarrow (\text{alphas\_cumprod}[\mathbf{t}^s]/\text{alphas\_cumprod}[\mathbf{t}^e])^{1/2}$
4: $\boldsymbol{\lambda}_s \leftarrow 1/\boldsymbol{\gamma}, \boldsymbol{\eta}_s \leftarrow -(1-\boldsymbol{\gamma}^2)^{1/2}/\boldsymbol{\gamma}$
5: denom $\leftarrow \boldsymbol{\lambda}_s \odot (\mathbf{t}_c - \mathbf{t}^s) + (\mathbf{t}^e - \mathbf{t}_c)$
6: $\boldsymbol{\lambda}_t \leftarrow \boldsymbol{\lambda}_s \odot (\mathbf{t}^e - \mathbf{t}^s)/\text{denom}$
7: $\boldsymbol{\eta}_t \leftarrow \boldsymbol{\eta}_s \odot (\mathbf{t}^e - \mathbf{t}_c)/\text{denom}$
8: $\mathbf{X}_{pred}^e \leftarrow \boldsymbol{\lambda}_t \odot \mathbf{X} + \boldsymbol{\eta}_t \odot \boldsymbol{\epsilon}_\theta$
9: $\mathbf{v}_\theta \leftarrow (\mathbf{X}_{pred}^e - \mathbf{X})/(\mathbf{t}^e - \mathbf{t}_c)$
10: $\mathbf{t}_{next} \leftarrow \text{GetNextTimesteps}(\mathbf{t})$
11: $\Delta\mathbf{t} \leftarrow (\mathbf{t}_{next} - \mathbf{t})/T_{max}$
12: $\mathbf{X}_{next} \leftarrow \mathbf{X} + \Delta\mathbf{t} \odot \mathbf{v}_\theta$
13: **return** $\mathbf{X}_{next}$

---

$i$-th denoising stage with timestep $t_i$. We maintain a latent buffer $\mathcal{B} = \{z_1^{(S_1)}, z_2^{(S_2)}, \ldots, z_{N-1}^{(S_{N-1})}\}$ storing intermediate latents at different stages.

At each iteration $j$, a new sample $z_{new}$ enters the pipeline at stage $S_0$. The pipeline batch is constructed as $\mathcal{P}_j = [z_{new}^{(S_0)}, z_1^{(S_1)}, \ldots, z_{N-1}^{(S_{N-1})}]$ with corresponding heterogeneous timesteps $\mathbf{t}_j = [t_0, t_1, \ldots, t_{N-1}]$. A single UNet forward pass processes all stages: $\boldsymbol{\epsilon}_\theta(\mathcal{P}_j, \mathbf{t}_j, c) = [\epsilon_0, \epsilon_1, \ldots, \epsilon_{N-1}]$ where $c$ is the conditioning.

### 3.4.2. ASYNCHRONOUS QUEUE PROCESSING

After the UNet forward pass, we apply the batched scheduler (Algorithm 1) to obtain updated latents $\mathcal{P}_j'$. The pipeline buffer is then updated via a shift operation: $\mathcal{B}_{j+1} = [z_{new}'^{(S_1)}, z_1'^{(S_2)}, \ldots, z_{N-1}'^{(S_{N-1})}]$. The completed sample $z_{N-1}'^{(S_N)}$ exits the pipeline for VAE (Kingma & Welling, 2022) decoding. This asynchronous processing ensures continuous throughput: once the pipeline is full (after $N - 1$ warmup iterations), each UNet call produces one complete sample.

### 3.4.3. THROUGHPUT ANALYSIS

Let $C_{UNet}$ denote the cost of a single UNet forward pass, $C_{VAE}$ the VAE decoding cost, and $C_{sched}$ the scheduler cost. For generating $M$ images, the vanilla approach requires $T_{vanilla} = M \cdot (N \cdot C_{UNet} + N \cdot C_{sched} + C_{VAE})$, while StreamFlow requires:

$$T_{ours} = (M + N - 1) \cdot C_{UNet} + (M + N - 1) \cdot C_{sched} + M \cdot C_{VAE} \approx M \cdot (C_{UNet} + C_{sched} + C_{VAE})$$
(7)

for $M \gg N$. The speedup factor is approximately $N$ when

---

**Algorithm 2** Pipeline Batch Denoising

---

**Require:** Conditioning $c$, number of images $M$, denoising steps $N$
**Ensure:** Generated images $\{I_1, \ldots, I_M\}$
1: Initialize: $\mathcal{B} \leftarrow \emptyset$, $\mathbf{t} \leftarrow [t_0, \ldots, t_{N-1}]$, initialized $\leftarrow$ False
2: **for** $j = 1$ to $M + N - 1$ **do**
3:    $z_{new} \sim \mathcal{N}(0, I); \mathcal{P}_j \leftarrow [z_{new}, \mathcal{B}]$
4:    **if** $|\mathcal{P}_j| = N$ and not initialized **then**
5:       initialized $\leftarrow$ True
6:    **end if**
7:    $\mathbf{t}_{batch} \leftarrow \mathbf{t}[0 : |\mathcal{P}_j|]$
8:    Apply CFG if needed: $\mathcal{P}_{in}, \mathbf{t}_{in}, c_{in} \leftarrow$ ApplyCFG($\mathcal{P}_j, \mathbf{t}_{batch}, c$)
9:    $\boldsymbol{\epsilon}_\theta \leftarrow \text{UNet}(\mathcal{P}_{in}, \mathbf{t}_{in}, c_{in})$
10:    $\boldsymbol{\epsilon}_\theta \leftarrow \text{HandleCFG}(\boldsymbol{\epsilon}_\theta)$
11:    $\mathcal{P}_j' \leftarrow \text{BatchVelocityStep}(\boldsymbol{\epsilon}_\theta, \mathcal{P}_j, \mathbf{t}_{batch})$
12:    **if** initialized **then**
13:       $I_{j-N+1} \leftarrow \text{VAE.decode}(\mathcal{P}_j'[-1]); \mathcal{B} \leftarrow \mathcal{P}_j'[0 : -1]$
14:    **else**
15:       $\mathcal{B} \leftarrow \mathcal{P}_j'$
16:    **end if**
17: **end for**
18: **return** $\{I_1, \ldots, I_M\}$

---

$C_{UNet}$ dominates. For $N = 4$ denoising steps, we achieve theoretical $4\times$ speedup. Algorithm 2 describes the complete pipeline batching procedure.

## 3.5. Dynamic TensorRT Compilation

### 3.5.1. CHALLENGE: STATIC COMPILATION VS. DYNAMIC STRUCTURE

TensorRT (NVIDIA Corporation, 2024) is a deep learning inference optimizer that compiles neural networks into optimized execution graphs. However, it assumes *static* inputs (Chen et al., 2018): all samples in a batch must have identical shapes and processing paths. Our heterogeneous timestep batching violates this assumption because different timesteps may trigger different execution branches (Shen et al., 2021) in the time-windows mechanism.

As shown in Fig. 3, formally, let $G_t$ denote the computational graph for timestep $t$. For a batch with heterogeneous timesteps $\mathbf{t} = [t_0, \ldots, t_{N-1}]$, the combined graph is $G_{batch} = \bigcup_{i=0}^{N-1} G_{t_i}$. If $G_{t_i} \neq G_{t_j}$ for some $i, j$, TensorRT compilation fails because it cannot determine a single static graph at compile time.

### 3.5.2. RUNTIME ADAPTIVE STRATEGY

We address this by introducing a runtime detection and decomposition mechanism. At each forward pass,

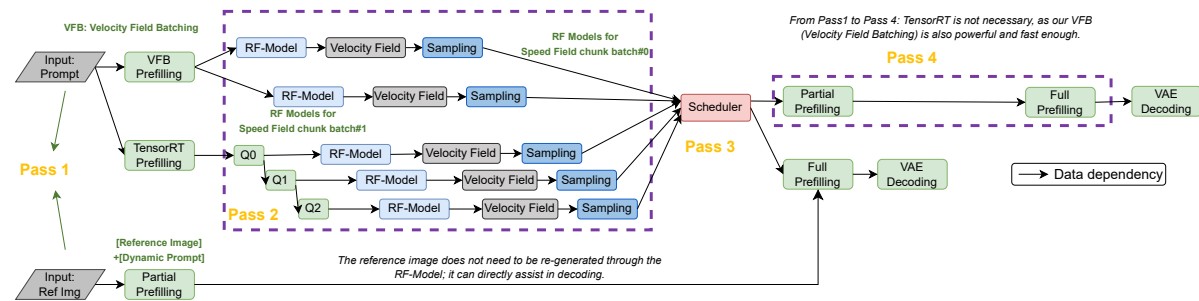

*Figure 3.* **Detailed implementation of StreamFlow's batched velocity field processing and heterogeneous timestep pipeline.** Starting from prompt and reference image inputs, **Pass 1** performs TensorRT-optimized/VFB (Velocity Field Batching) prefilling to establish initial latent representations. **Pass 2** demonstrates the core velocity field batch processing: multiple RF model instances process parallel queues (Q0, Q1, Q2) with heterogeneous timesteps, where each queue independently computes velocity fields before synchronization through reranking modules. **Pass 3** shows the asynchronous queue management: the unified adaptive scheduler coordinates multiple parallel streams, each progressing through partial prefilling → full prefilling → VAE decoding stages at different rates. **Pass 4** illustrates the dynamic dependency graph that enables proper data flow across heterogeneous processing stages. Data dependency arrows show how velocity field outputs from Pass 2 feed into the scheduler, which then manages the asynchronous progression of multiple generation streams in Pass 3, achieving continuous throughput without blocking.

we check whether the timestep batch is homogeneous: $\text{is\_homogeneous}(\mathbf{t}) = \mathbb{I}[t_i = t_j \ \forall i, j]$.

**Case 1: Homogeneous batch.** All timesteps are identical. We directly invoke the compiled TensorRT engine: $\epsilon_\theta = \text{TensorRT}(\mathcal{P}, t, c)$. This is the common case during warmup and when using batch generation without pipeline.

**Case 2: Heterogeneous batch.** Timesteps differ. We decompose the batch into individual samples: $\epsilon_\theta = \bigoplus_{i=0}^{N-1} \text{TensorRT}(z_i, t_i, c_i)$ where $\bigoplus$ denotes concatenation along the batch dimension. While this introduces $N$ TensorRT calls instead of 1, the per-call overhead is minimal ($<$1ms), and the overall speedup from TensorRT optimization (typically $2-3\times$) still provides substantial gains.

**Performance trade-off.** Let $O_{TRT}$ be the TensorRT overhead per call. The total cost becomes $C_{heterogeneous} = N \cdot (O_{TRT} + C_{UNet}^{TRT}) \ll N \cdot C_{UNet}^{vanilla}$. Since $C_{UNet}^{TRT} \approx 0.3 \cdot C_{UNet}^{vanilla}$ and $O_{TRT} \approx 0.001 \cdot C_{UNet}^{vanilla}$, we still achieve $\approx 3\times$ speedup despite the decomposition.

Algorithm 3 presents the complete adaptive TensorRT forward pass.

**Plug-and-play compatibility.** Our approach wraps the TensorRT engine without modifying the compilation process itself. This ensures compatibility with any TensorRT-compiled UNet, requiring zero changes to existing acceleration pipelines. The runtime overhead of homogeneity checking is negligible ($<$0.01ms).

## 4. Experiments

Our work differs from previous acceleration frameworks which focus on optimizing different models(Ma et al., 2023; Li et al., 2024). Therefore, they do not compete with each other. Instead, our approach is compatible with the ex-

---

**Algorithm 3** TensorRT Compatible Forward

---

**Require:** Latents $\mathcal{P}$, timesteps $\mathbf{t}$, conditioning $c$
**Ensure:** Model output $\epsilon_\theta$
1: $\mathbf{t}_{unique} \leftarrow \text{Unique}(\mathbf{t})$
2: **if** $|\mathbf{t}_{unique}| = 1$ **then**
3:      $\epsilon_\theta \quad\quad\leftarrow\quad\quad \text{TensorRT\_Forward}(\mathcal{P}, \mathbf{t}[0], c)$ {Homogeneous case}
4: **else**
5:      results $\leftarrow [\text{TensorRT\_Forward}(\mathcal{P}[i], \mathbf{t}[i], c[i])$ for $i \in [0, B-1]]$ {Heterogeneous decomposition}
6:      **if** isinstance(results[0], tuple) **then**
7:          $\epsilon_\theta \leftarrow \text{ConcatenateTuples}(\text{results})$
8:      **else**
9:          $\epsilon_\theta \leftarrow \text{Concatenate}(\text{results}, \dim = 0)$
10:      **end if**
11: **end if**
12: **return** $\epsilon_\theta$

---

isting Rectified Flow models' acceleration methods[2](Liu et al., 2024; Yan et al., 2024). They can be used in combination rather than being mutually exclusive. Therefore, we mainly focused on the effectiveness and reliability of our method, and conducted detailed ablation evaluations, power and memory efficiency evaluations, scalability robustness evaluations, and quality evaluations, etc. We test our pipeline on a workstation equipped with 8x NVIDIA Quadro RTX 8000 GPUs (46GB VRAM each), Intel Xeon Silver 4116 CPU @ 2.10GHz, and Ubuntu 24.04 LTS for image generation. We measured the inference time for an average of 100 images, following the convention of previous studies (Kodaira et al., 2025; Feng et al., 2025).

---

[2]Accelerated methods for the traditional Rectified Flow model are generally non-transferable, require architecture changes, re-training or distillation.

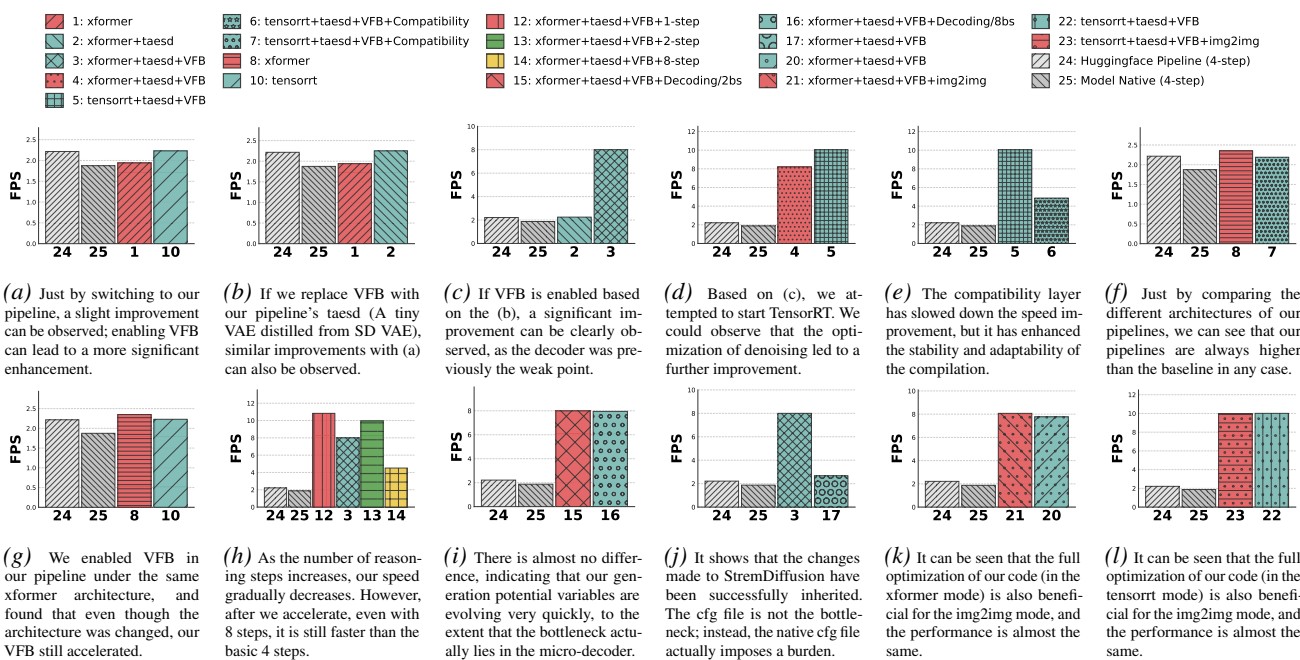

*(a)* Just by switching to our pipeline, a slight improvement can be observed; enabling VFB can lead to a more significant enhancement.

*(b)* If we replace VFB with our pipeline's taesd (A tiny VAE distilled from SD VAE), similar improvements with (a) can also be observed.

*(c)* If VFB is enabled based on the (b), a significant improvement can be clearly observed, as the decoder was previously the weak point.

*(d)* Based on (c), we attempted to start TensorRT. We could observe that the optimization of denoising led to a further improvement.

*(e)* The compatibility layer has slowed down the speed improvement, but it has enhanced the stability and adaptability of the compilation.

*(f)* Just by comparing the different architectures of our pipelines, we can see that our pipelines are always higher than the baseline in any case.

*(g)* We enabled VFB in our pipeline under the same xformer architecture, and found that even though the architecture was changed, our VFB still accelerated.

*(h)* As the number of reasoning steps increases, our speed gradually decreases. However, after we accelerate, even with 8 steps, it is still faster than the basic 4 steps.

*(i)* There is almost no difference, indicating that our generation potential variables are evolving very quickly, to the extent that the bottleneck actually lies in the micro-decoder.

*(j)* It shows that the changes made to StremDiffusion have been successfully inherited. The cfg file is not the bottleneck; instead, the native cfg file actually imposes a burden.

*(k)* It can be seen that the full optimization of our code (in the xformer mode) is also beneficial for the img2img mode, and the performance is almost the same.

*(l)* It can be seen that the full optimization of our code (in the tensorrt mode) is also beneficial for the img2img mode, and the performance is almost the same.

*Figure 4.* **Ablation Study:** We conducted a detailed ablation study on all the components, and the experiments proved that all the components have a significantly higher average generation speed compared to the original model or the official pipeline of Huggingface. Additionally, in the fastest case, our peak improvement can reach 11/1.8, which is approximately a 611% increase from the original speed.

*Table 1.* **Power/Memory Comparison:** Higher FPS is better; lower power and peak memory are better. The img2img mode here is a noise-free mode because there is a reference image.

| Method | FPS ↑ | Power (W) ↓ | Peak Mem (MB) ↓ |
|---|---|---|---|
| xformer+taesd+VFB+1-step | 10.83 | 171.95 | 2976 |
| tensorrt+taesd+VFB | 10.05 | 110.51 | 2666 |
| tensorrt+taesd+VFB+img2img | 10.01 | 104.50 | 2694 |
| xformer+taesd+VFB | 8.00 | 153.17 | 3012 |
| xformer+taesd+VFB+img2img | 7.77 | 31.82 | 2966 |
| Baseline HF (von Platen et al., 2022) | 2.22 | 196.68 | 3696 |
| Baseline Vanilla (Yan et al., 2024) | 1.87 | 205.28 | 3788 |

### 4.1. Ablation Assessment

Our framework inherits two architectural-level optimizations from Xformer(Zhang et al., 2024) and TensorRT (NVIDIA Corporation, 2024). These can be switched, so there are two scenarios that need to be compared. Without any special indication, all are four-step inference speeds. VFB refers to Velocity Field Batching; taesd refers to the Tiny VAE (Kingma & Welling, 2022) included in the acceleration framework; By default, all images are set to a size of 512×512.

As demonstrated in Figure 4, we conducted comprehensive ablation studies to validate each component's contribution. The baseline performance shows that the vanilla implementation achieves 1.87 FPS while the Hugging Face pipeline reaches 2.22 FPS. When we incrementally add optimizations starting with Xformer architecture (Fig 4a) and taesd decoder (Fig 4b), we observe modest improvements

to 2.3 FPS, confirming that tiny VAE decoding represents a significant bottleneck(Rombach et al., 2022b). The most substantial gains emerge when enabling VFB with taesd (Fig 4c), achieving 8.0 FPS—a 360% improvement that validates our core insight that batched velocity field (Lipman et al., 2023) processing is essential for Rectified Flow models. Further enabling TensorRT compilation (Fig 4d) pushes performance to 10.05 FPS, while adding the compatibility layer (Fig 4e) introduces a minor trade-off (9.8 FPS) but enhances stability across diverse architectures. Our experiments reveal important scalability insights: even with 8 inference steps, our optimized pipeline (4.5 FPS, Fig 4h) outperforms the 4-step baseline (2.22 FPS), and the decoder batch size comparison (Fig 4i) shows that latent generation is no longer the bottleneck—the system is now limited by VAE decoding throughput.

### 4.2. Efficiency Assessment

Beyond raw speed improvements, we evaluate the resource efficiency of StreamFlow across power consumption and memory footprint metrics. Table 1 presents comprehensive measurements for our key configurations alongside baseline methods. The tensorrt+taesd+VFB setup achieves 10.05 FPS while consuming only 104-110W, compared to 205W for the vanilla baseline—representing a 47% reduction in power consumption while delivering 5.4× speedup, translating to approximately 11× better performance per watt (Schwartz et al., 2019). Even Xformer-based

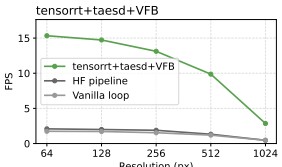

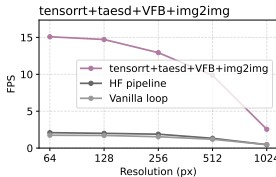

(*a*) Our method can still function after being compiled by TensorRT. However, it will be affected when the image size increases, but the baseline is more significantly impacted.

(*b*) When in the img2img mode, we can observe that this has very little impact, which is also consistent with the performance of the ablation study.

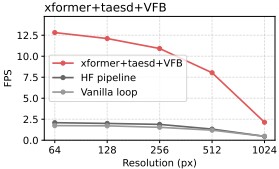

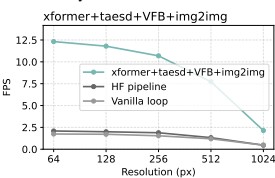

(*c*) Under the xformer framework, our initial acceleration ratio is slightly lower than that of TensorRT, but the reduction in our ratio is slightly less than TensorRT.

(*d*) Then we can observe that the accelerated framework in the xformer mode performs quite robustly. Even if the baseline is significantly affected, we do not suffer much loss.

*Figure 5.* **Scalability Study:** By changing the target size of the generated images, we found that our method has much stronger robustness compared to the previous methods. We are hardly affected by too much size variation. Even at relatively large sizes, we still achieve an acceleration of almost four to five times compared to the baseline. This not only demonstrates the robustness of our method, but also proves that our approach is a true method-to-strategy acceleration, rather than taking any expedient path.

configurations demonstrate favorable power profiles, with xformer+taesd+VFB consuming 153W at 8.0 FPS versus the baseline's 196W at 2.22 FPS. Memory efficiency constitutes another critical advantage of StreamFlow. Our optimized pipelines reduce peak memory usage from 3788MB (vanilla baseline) to 2666-3012MB across various configurations, achieving 20-30% memory savings through our batched velocity field processing strategy, which eliminates redundant intermediate tensor allocations. The TensorRT configurations achieve the lowest memory footprint (2666-2694MB) due to aggressive operator fusion and memory planning (Chen et al., 2018; Dao et al., 2022). Combined with throughput improvements, our framework enables serving 5-6× more requests per GPU, dramatically improving infrastructure utilization and cost efficiency in production environments.

### 4.3. Scalability Assessment

To evaluate the robustness of our acceleration framework under varying computational demands, we conduct a scalability study across different image resolutions ranging from 64px to 1024px. As illustrated in Figure 5, our method demonstrates remarkable stability compared to baseline approaches, maintaining substantial speedup ratios even as resolution increases. Under TensorRT compilation (Fig. 5a, 5b), our tensorrt+taesd+VFB configuration achieves consistent 4-5× acceleration across all resolutions in both text-to-image and img2img modes. The Xformer-based con-

*Table 2.* **Quality Study:** Higher ClIP score is better; lower FID are better. It can be seen that our acceleration scheme has almost no impact on the generation quality (the impact is less than 1%).

| Method | ClIP score ↑ | FID ↓ | VAE |
|---|---|---|---|
| tensorrt+taesd+VFB | 96.25 | 31.28 | taesd |
| tensorrt+Original VAE+VFB | 96.08 | 31.16 | Original VAE |
| xformer+taesd+VFB | 97.26 | 30.75 | taesd |
| xformer+Original VAE+VFB | 96.74 | 31.82 | Original VAE |
| Baseline HF (von Platen et al., 2022) | 97.27 | 30.08 | Original VAE |
| Baseline Vanilla (Yan et al., 2024) | 97.28 | 29.44 | Original VAE |

configurations (Fig. 5c, 5d) exhibit even stronger scalability—at 1024px resolution, while baseline methods experience severe performance collapse, our xformer+taesd+VFB configuration maintains nearly 80% of its peak throughput. This resilience stems from our decoupled velocity field processing, which adapts naturally to increased tensor dimensions without introducing architectural bottlenecks, validating that StreamFlow provides genuine algorithmic-level acceleration rather than exploiting resolution-specific optimizations.

### 4.4. Quality Assessment

To verify that our acceleration framework does not compromise generation quality, we evaluate CLIP Score and FID on standard text-to-image benchmarks. As shown in Table 2, our method maintains generation quality comparable to the baselines, with less than 1% variation in CLIP Score. These results confirm that StreamFlow's core acceleration mechanisms preserve generation fidelity while delivering substantial speedups, making our framework suitable for production deployment where both speed and quality are critical.

## 5. Conclusion

In this paper, we present the first comprehensive acceleration framework specifically designed for accelerating the Rectified Flow model. We have thoroughly analyzed the difficulties (Velocity Field Batching, Heterogeneous Timesteps, Dynamic Compilation) in designing a new acceleration framework for flow models from a theoretical, strategic, and practical perspective. We have also understood the reasons for the failure of traditional methods and proposed corresponding solutions. Our acceleration framework is plug-and-play and can significantly increase the generation speed of 512-dimensional images of the Rectified Flow model by 611% with almost no loss in quality. Even on larger dimensions, we can still achieve an almost 4-5 times improvement, and in this case, the native Hugging Face pipeline is almost unable to accelerate. Compared with the inconvenient-to-reuse methods such as distillation, Reflow, and few-step, we directly make the deployment of large-scale flow models possible.

## Impact Statement

This paper presents work whose goal is to advance the field of Machine Learning. There are many potential societal consequences of our work, none which we feel must be specifically highlighted here.

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
