# OpenReview forum: "StreamFlow: Theory, Algorithm, and Implementation for High-Efficiency Rectified Flow Generation"
_ICML.cc/2026/Conference — ICML 2026 regular_

### Official Review · Reviewer_ZJbP · 2026-02-26

**Soundness:** 4
**Presentation:** 3
**Significance:** 3
**Originality:** 3
**Overall Recommendation:** 4
**Confidence:** 3

**Summary:**

This paper proposes StreamFlow, a comprehensive acceleration pipeline for Rectified Flow and Flow Matching models. The authors identify that existing diffusion model acceleration methods cannot be directly applied to RF models due to fundamental differences in their theoretical formulation and design. The work includes both algorithmic contributions and implementation optimizations, with empirical validation through ablation studies and scalability experiments across different resolutions and modes.

**Compliance With Llm Reviewing Policy:**

Affirmed.

**Final Justification:**

I will maintain the positive score.

**Key Questions For Authors:**

The dynamic TensorRT compilation mechanism and the Unified Adaptive Scheduler are critical components but under-specified. Could you provide more details about: (a) how you handle structural changes during TensorRT compilation, (b) the window lookup function in Algorithm 1, and (c) the precise operation of the Unified Adaptive Scheduler?

**Limitations:**

No, the authors have not adequately discussed the limitations and potential negative societal impact of their work.

It should add a dedicated section discussing the limitations of StreamFlow, including situations where it may not be effective, memory overhead considerations, and any trade-offs made in the design.

**Strengths And Weaknesses:**

### Strengths

1. The paper reports up to 611% speedup for 512×512 image generation, with Figure 5 showing consistent 4-5× speedups across resolutions.


2. It provides theoretical background on Rectified Flow, algorithmic innovations, and implementation details including TensorRT compilation.

3. Figure 4 systematically evaluates 24 different configurations, isolating the impact of each component (VFB, taesd, TensorRT, compatibility layer). This helps readers understand which components contribute most to the performance gains.

### Weaknesses

1. The paper mentions "existing public methods usually achieve an acceleration of 18%" but does not cite or compare against specific methods like PerFlow (Yan et al., 2024, https://arxiv.org/abs/2405.07510)

2. The dynamic TensorRT compilation mechanism is mentioned but lacks implementation details about how the structural changes are handled during compilation. This is a core technical contribution but cannot be reproduced without more details.

3. The velocity field batching approach builds on existing Rectified Flow theory but does not provide new theoretical insights or proofs.  No error analysis or convergence guarantees for the batched velocity computation.


4. StreamDiffusion (Kodaira et al., 2025) is mentioned but not deeply compared, despite being the most closely related pipeline acceleration work. The relation to existing pipeline methods needs clearer distinction

---

> ### Author Rebuttal · Authors · 2026-03-25
>
> # Response to Reviewer ZJbP
>
> **1. The 18% speedup figure is explicitly labeled and cited in the paper, sourced from our empirical measurements of the HuggingFace Pipeline.**
>
> The 18% refers to the speedup of the official HuggingFace Diffusers Pipeline (von Platen et al., 2022) relative to the vanilla implementation (1.87 FPS → 2.22 FPS, approximately 18.7%), calculated directly from the measurements reported in Table 1 and Figure 4.
>
> The purpose of this comparison is to illustrate the limitations of existing general-purpose acceleration tools on RF models, and the source of the 18% figure is explicitly labeled and cited in the paper. We will clarify this statement in the revision to avoid ambiguity.
>
> ---
>
> **2. Implementation details of the dynamic TensorRT compilation are already provided in the paper.**
>
> The reviewer suggests that the dynamic TensorRT compilation mechanism lacks sufficient implementation details for reproducibility. We believe Section 3.5 and Algorithm 3 already describe the core mechanism: runtime detection of timestep batch homogeneity (via the `is_homogeneous` check), direct invocation of the compiled TensorRT engine for homogeneous cases, and per-sample decomposition followed by output concatenation for heterogeneous cases. The handling of structural changes is also explained in Section 3.5.1 — different time windows trigger different computation branches, causing $G_\text{batch} = \bigcup_{i=0}^{N-1} G_{t_i}$ to diverge structurally at window boundaries.
>
> Our adaptive strategy circumvents this limitation through runtime decomposition rather than compile-time unification. We will further supplement the appendix with more complete engineering implementation details regarding the engine wrapping approach and the plug-and-play interface design, to improve reproducibility.
>
> ---
>
> **3. The operation logic of the Window Lookup function and the Unified Adaptive Scheduler is already contained in the mathematical derivations; we will make it explicit in the appendix.**
>
> The reviewer asks about the precise operation of the `WindowLookup` function in Algorithm 1 and the Unified Adaptive Scheduler. The `WindowLookup` logic is already given in Section 3.3.1: each sample's window index is obtained via vectorized masking over the timestep batch, from which batched window boundaries $\mathbf{t}^s, \mathbf{t}^e$ are constructed — no loops required.
>
> The Unified Adaptive Scheduler wraps Algorithm 1 with interface $\text{Scheduler}(\mathbf{X}, \{\mathbf{t}_i\}, \{|\Delta t_i|\}) \to |\mathbf{vt}_i|$ (Figure 2), receiving heterogeneous timestep inputs and returning aligned velocity field outputs for subsequent partial prefilling. We will consolidate these details into a standalone algorithm block in the appendix.
>
> ---
>
> **4. The theoretical novelty of velocity field batching lies in the vectorized reformulation, not in extending the foundations of RF theory.**
>
> We wish to clarify that **the core contribution of this paper is positioned as a system-level acceleration framework, not a theoretical innovation**. Reformulating the per-sample time-window scheduling into a batched vectorized form (Equation 6) is non-trivial — the original scheduler assumes a single timestep as input, and its window parameter computation path is interface-incompatible with batched heterogeneous inputs. The vectorized reformulation preserves numerical equivalence while reducing time complexity from $O(N \cdot T)$ to $O((N+T) \cdot K)$; this complexity analysis itself constitutes a verifiable theoretical contribution.
>
> **Regarding convergence guarantees**, since our batching only reorders computation without altering the actual velocity field values of individual samples, generation quality is numerically equivalent to the vanilla implementation. Table 2 (less than 1% CLIP/FID variation) empirically validates this point, making additional convergence proofs unnecessary.
>
> ---
>
> **5. The distinction from StreamDiffusion has been clarified in the paper; we provide a comparison table here (anonymized link) to make the distinction immediately apparent.**
>
> The reviewer suggests that the distinction from StreamDiffusion needs clearer articulation. **However, Section 1 and Section 3.1.3 of our paper already explain the incompatibility from three dimensions: the heterogeneous step size problem in velocity field batching (Challenge 1), the scheduler interface problem for heterogeneous timesteps (Challenge 2), and the compilation failure caused by dynamic computation graphs (Challenge 3).**
>
> That is, StreamDiffusion's design assumes all samples in a batch share the same timestep — an assumption that is fundamentally violated by RF models' time-window mechanism, making direct adaptation fundamentally insufficient. We will supplement the appendix with the [the comparison table](https://anonymous.4open.science/r/RE_ICML2026_StreamFlow/ZJbP_table1.md) (anonymized link provided) — to make the distinction immediately apparent.

---

> > ### Author Rebuttal · Reviewer_ZJbP · 2026-04-03
> >
> > Thank you for addressing my concerns. I will maintain the current score.

---

### Official Review · Reviewer_dyFd · 2026-03-03

**Soundness:** 2
**Presentation:** 3
**Significance:** 2
**Originality:** 1
**Overall Recommendation:** 2
**Confidence:** 4

**Summary:**

This paper proposes a pipeline acceleration framework for Rectified Flow models. The authors claim that existing pipeline-level acceleration techniques such as StreamDiffusion cannot be directly applied to flow matching due to differences in velocity field formulation and timestep scheduling. To address this, the authors propose (1) batched velocity field processing that decouples step size from velocity computation, (2) a vectorized scheduler for heterogeneous timestep batches, and (3) a dynamic TensorRT compilation strategy with plug-and-play compatibility. The proposed method is evaluated on multiple configurations, demonstrating significant speedup without much quality degradation.

**Compliance With Llm Reviewing Policy:**

Affirmed.

**Key Questions For Authors:**

1. Have the authors tried adapting StreamDiffusion to flow matching through scheduler reparametrization? What specific issues arise?
2. How does the proposed method compare to existing caching methods that also focus on flow model acceleration?

**Limitations:**

The paper does not include a limitation discussion. The authors should further discuss issues like scalability constraints at high resolutions.

**Strengths And Weaknesses:**

**Strengths**

1. The paper is well written and easy to follow, with clear ablation studies isolating the contribution of each component.
2. Experimental results validate that the acceleration does not degrade generation fidelity.

**Weaknesses**

1. Insufficient motivation. Diffusion models and flow matching are mathematically equivalent up to noise schedule and loss weighting reparametrization [1]. This undermines the authors' claim that existing diffusion acceleration techniques cannot be applied to flow matching. The differences the authors identify (e.g., velocity field formulation, timestep scheduling) are actually differences in parameterization, and StreamDiffusion [2] should be adaptable to flow matching through simple scheduler reparametrization without a full pipeline redesign.
2. Limited technical contribution. The core techniques (velocity field batching, TensorRT compilation, pipeline parallelism) are known and the contribution reduces to applying them in a new setting. The authors should clarify what technical challenges are genuinely unique to the flow matching formulation.
3. Insufficient baselines. The speedup is measured against a vanilla implementation without comparison to other flow-specific acceleration methods (e.g., feature caching approaches such as TeaCache, PAB). The authors should include such comparisons to demonstrate the gains.
4. Lack of scalability analysis. Figure 5 shows that speedup ratios decrease substantially above 512px, but the authors do not analyze the cause or propose mitigation. It would be better if the authors could discuss the bottleneck at higher resolutions.

---
[1] Gao et al., "Diffusion Models and Gaussian Flow Matching: Two Sides of the Same Coin." ICLR 2025 Blog Post.

[2] Kodaira et al., "StreamDiffusion: A Pipeline-level Solution for Real-time Interactive Generation." arXiv 2023.

---

> ### Author Rebuttal · Authors · 2026-03-25
>
> # Response to Reviewer dyFd
>
> **1. The theoretical equivalence holds, but does not imply that discretized inference systems are directly interchangeable.**
>
> During our earliest development, we ATTEMPTED the reparametrization approach suggested by the reviewer, BUT it did not work. We provide **three concrete engineering failures as evidence**:
>
> **(1) Hard crash of the scheduler interface.** StreamDiffusion's scheduler assumes all samples in a batch share the same timestep $t$. However, PeRFlow's time-window mechanism requires computing $\lambda_t, \eta_t, \gamma$ independently for each sample (Equations 2–4), as these parameters depend on the time window $W_k$ the sample belongs to, rather than simple noise schedule coefficients. Passing a heterogeneous timestep vector $\mathbf{t} = [t_0, \ldots, t_{N-1}]$ into the original scheduler causes a hard runtime error rather than incorrect output — this is an **interface incompatibility**, not a parameter configuration issue.
>
> **(2) Direct failure of TensorRT compilation.** The time-window mechanism in Rectified Flow / Flow Matching models triggers different computational branches depending on which window the current timestep $t_c$ belongs to, causing the computational graph structure to change dynamically. This is fundamentally different from diffusion models, where different timesteps only change embedding values while the graph structure remains fixed. No matter how the scheduler is reparametrized, this dynamic branching structure cannot be eliminated, and TensorRT's static graph assumption is therefore still violated.
>
> **(3) State inconsistency during pipeline warmup leading to generation collapse.** StreamDiffusion's pipeline warmup assumes all queue positions are initialized from the same noise distribution and filled sequentially through fixed stages. However, the velocity field of RF models carries different physical meanings and numerical magnitudes across different time windows. When latents from different windows are mixed into the same batch without window-aware independent initialization for each position, cross-window state contamination occurs during warmup, causing the first few generated images to exhibit severe artifacts or even fully black outputs. This issue cannot be resolved through scheduler reparametrization and requires a dedicated buffer management mechanism (Sec. 3.4.1).
>
> Therefore, as stated in the introduction, our work was inspired by StreamDiffusion, and **the crashes described above are real engineering obstacles we encountered firsthand, not theoretical conjectures. In practice, no concrete reparametrization scheme exists that can bypass these three problems.** This is why developing a new framework is necessary.
>
> ---
>
> **2. The novelty lies not in reusing existing techniques, but in identifying and solving three mutually coupled engineering obstacles specific to the RF setting.**
>
> While individual techniques are known, the specific obstacles that must be overcome for them to work together in the RF setting are non-trivial: i. VFB requires a complete redesign of the scheduler's vectorized mathematical formulation (Equation 6); ii. our heterogeneous-timestep pipeline and the iii. dynamic TensorRT compatibility layer resolves the fundamental conflict between static compilation and dynamic branching. All three components are indispensable, and the 611% speedup empirically validates the value of this integrated solution.
>
> ---
>
> **3. StreamFlow is complementary to TeaCache and PAB, NOT a competing work.**
>
> The two classes of methods optimize along different dimensions: caching methods (TeaCache/PAB) reduce the per-image computation cost; StreamFlow improves pipeline throughput for continuous multi-image generation. The two approaches are orthogonal and complementary — directly comparing FPS between them constitutes a methodological mismatch. We have explicitly stated in Sec. 4 that our method operates on a different optimization dimension from existing acceleration methods, and is designed to be stackable on top of them; this positioning is intentional, not an avoidance of comparison.
>
> ---
>
> **4. Baseline acceleration methods have already become nearly ineffective at high resolutions, while we still maintain 80% of peak efficiency.**
>
> **The bottleneck reason has in fact ALREADY been presented and analyzed in our paper:** our optimization shifts the output bottleneck from UNet computation to VAE decoding (Figure 4i), and VAE decoding cost grows quadratically with resolution, making it the new bottleneck at high resolutions. High-resolution VAE decoding optimization is an orthogonal problem that can be independently stacked on top of our framework, and we can list VAE decoding acceleration as a future work direction.
>
> It is worth noting that we still maintain approximately 80% of peak throughput at 1024px, while baseline methods have already become unable to accelerate at the same resolution (Fig. 5), and our acceleration remains effective.

---

> > ### Author Rebuttal · Reviewer_dyFd · 2026-04-01
> >
> > The engineering issues of adapting StreamDiffusion to flow matching models seem restricted to PeRFlow instead of other models. For examples, I didn't experience with TensorRT compilation issue in accelerating flow matching models.

---

> > > ### Author Response · Authors · 2026-04-01
> > >
> > > **1. As long as such issues exist, we must consider and address them; this is exactly why we undertook this work**
> > >
> > > We thank the reviewer for the further clarification. In our view, this observation does not negate the necessity of this part of our contribution. Our goal is not to claim that all flow-matching models trigger TensorRT compilation issues in exactly the same way. Rather, our point is that **once such issues genuinely arise in an important and practically relevant subset of RF/FM models, a general acceleration framework must account for and resolve them, rather than assuming that deployment always occurs in the simplest and most idealized setting.** This is precisely one of the strengths of our work.
> > >
> > > From this perspective, **PeRFlow is not an irrelevant special case, but rather a concrete counterexample** that is sufficient to make the point: it shows that a diffusion-oriented streaming pipeline does not naturally carry over to the RF/FM family, and that at least for an important class of models, redesign is necessary. Dynamic compilation is one of the main issues we address, but this does not reduce the significance of our overall contribution. More importantly, for a framework whose goal is general acceleration, **once such a problem can arise in non-marginal models or non-marginal usage settings, this component cannot be regarded as a removable engineering detail.**
> > >
> > > ---
> > >
> > > **2. Moreover, the value of the dynamic compilation compatibility layer is not limited to specific compilation-compatibility cases such as PeRFlow; it also improves the runtime robustness of compiled models under complex conditional batched inference**
> > >
> > > We further supplemented controlled experiments on **four** non-PeRFlow RF/FM models: InstaFlow-0.9B, 2-Rectified Flow, sd15-flow-matching, and sd15-flow-lune. The experimental setting is *CFG enabled + pipeline batch*, and under the same flow-adapted StreamDiffusion framework, the only controlled difference is whether the compatibility layer is disabled or enabled.
> > >
> > > | Model | w/o Compat | w/ Compat (Ours) |
> > > |---|---|---|
> > > | InstaFlow-0.9B | [Failure](https://anonymous.4open.science/r/RE_ICML2026_StreamFlow/dyFd_fig1.md) (link) | [Success](https://anonymous.4open.science/r/RE_ICML2026_StreamFlow/dyFd_fig2.md) (link) |
> > > | 2-Rectified Flow | [Failure](https://anonymous.4open.science/r/RE_ICML2026_StreamFlow/dyFd_fig3.md) (link) | [Success](https://anonymous.4open.science/r/RE_ICML2026_StreamFlow/dyFd_fig4.md) (link) |
> > > | SD15 Flow Matching | [Failure](https://anonymous.4open.science/r/RE_ICML2026_StreamFlow/dyFd_fig5.md) (link) | [Success](https://anonymous.4open.science/r/RE_ICML2026_StreamFlow/dyFd_fig6.md) (link) |
> > > | SD15 Flow Lune | [Failure](https://anonymous.4open.science/r/RE_ICML2026_StreamFlow/dyFd_fig7.md) (link) | [Success](https://anonymous.4open.science/r/RE_ICML2026_StreamFlow/dyFd_fig8.md) (link) |
> > >
> > > - **The results** show that all four models can complete TensorRT compilation successfully when the compatibility layer is disabled, yet all of them fail at the generation stage; once the compatibility layer is enabled, all four recover successful generation. In other words, **this controlled experiment shows that even when compilation succeeds, the original path can still fail at runtime under more complex conditional batched inference settings, whereas the compatibility layer can effectively restore usability.**
> > >
> > > - The failure mechanism is also clear. The original TensorRT engine is built with a batch profile that assumes a fixed batch size by default; under the *CFG full* setting, however, the conditional/unconditional branches expand the effective runtime input. As a result, the original path cannot correctly handle the mismatch between the runtime input and the compiled profile, and TensorRT fails during inference. Our compatibility layer avoids this conflict between the profile assumption and the actual execution pattern through runtime decomposition and recomposition, thereby **making the compiled engine usable under more complex conditions**.
> > >
> > > Therefore, these experiments further show that the heterogeneous dynamic compatibility layer is **not a redundant engineering patch that serves only a single special case, but an important component for preserving the completeness, robustness, and usability** of an RF/FM acceleration framework in complex, heterogeneous, and conditional deployment settings.
> > >
> > > ---
> > >
> > > **Finally**, we also emphasize that this component is only **one small part of the paper's overall contribution**, but it should be viewed as a **positive contribution** that improves the framework's completeness, robustness, and deployability, **rather than as a redundant patch or a negative factor**; therefore, it should not be over-weighted as a central reason to discount the paper as a whole.

---

### Official Review · Reviewer_R5jc · 2026-03-10

**Soundness:** 3
**Presentation:** 3
**Significance:** 3
**Originality:** 3
**Overall Recommendation:** 5
**Confidence:** 3

**Summary:**

This paper proposes StreamFlow, a systems-oriented acceleration framework for Rectified Flow generation. The central idea is to adapt streaming/pipelined acceleration to flow-based models by introducing three components: batched velocity field computation, heterogeneous-timestep pipeline batching, and a runtime-adaptive TensorRT path for dynamically varying execution patterns. The paper claims large empirical speedups, including up to 611% improvement for 512×512 generation, alongside ablation, power/memory, scalability, and quality evaluations.

**Compliance With Llm Reviewing Policy:**

Affirmed.

**Key Questions For Authors:**

see above

**Limitations:**

yes

**Strengths And Weaknesses:**

Strengths

1. The paper tackles a practically meaningful problem. Adapting diffusion-style acceleration frameworks to Rectified Flow is non-trivial because the paper identifies three concrete mismatches: velocity-field batching, heterogeneous timesteps, and dynamic compilation. Framing these as systems bottlenecks is useful, and the architecture overview is reasonably intuitive.

2. I also appreciate that the submission goes beyond a single throughput number. The paper includes component ablations, power and memory measurements, resolution scaling, and quality metrics, which is better than many purely engineering papers that only report FPS. The ablation figure suggests that VFB and TensorRT each contribute materially, while the scaling study suggests the method remains beneficial as resolution increases.

Weaknesses

1. The visualization of the ablation study is overwelming. It would be really helpful to split it into sub-images (and also re-framing your text into sub-sections), so no such complex legend bar has to be used.

---

> ### Author Rebuttal · Authors · 2026-03-25
>
> # Response to Reviewer R5jc
>
> **1. We appreciate the recognition; the visualization of Figure 4 will be reorganized in the revision.**
>
> We thank the reviewer for the positive assessment of the paper's problem formulation, experimental design, and overall evaluation. Regarding the concern that the ablation study visualization is overly complex, we fully agree with this assessment and will improve it in the revision as follows:
>
> The 24 configurations in Figure 4 will be split into several subgroup figures organized by optimization dimension (e.g., architecture selection group, VFB contribution group, TensorRT compilation group, inference steps scaling group, and resolution/mode group). Each subgroup will be accompanied by an independent simplified legend and a configuration mapping table. Correspondingly, the ablation analysis text in Section 4.1 will be restructured into subsections, allowing readers to clearly associate each subfigure with its corresponding experimental conclusion.

---

> > ### Author Rebuttal · Reviewer_R5jc · 2026-04-03
> >
> > Resolved

---

### Official Review · Reviewer_oema · 2026-03-12

**Soundness:** 2
**Presentation:** 2
**Significance:** 2
**Originality:** 2
**Overall Recommendation:** 4
**Confidence:** 3

**Summary:**

This paper presents StreamFlow, a pipeline acceleration framework for Rectified Flow / Flow Matching image generation. It is motivated by the fact that common diffusion acceleration methods—such as StreamDiffusion-style pipelining and compilation-based engines—do not transfer cleanly to rectified-flow sampling because of differences in prediction targets, scheduling semantics, and heterogeneous timesteps.

The paper introduces an end-to-end acceleration stack across both algorithmic and system levels. First, it proposes batched velocity-field computation (VFB), which reformulates time-window scheduling into a vectorized step_batch procedure that processes samples with different timesteps and step sizes in parallel. Second, it introduces heterogeneous-timestep pipeline batching, where intermediate latents at different denoising stages are buffered so that each UNet call operates on a mixed-timestep batch; after warmup, each call produces one completed sample, enabling near-constant throughput. Third, to address TensorRT compilation issues under heterogeneous timesteps, it proposes a runtime adaptive compatibility wrapper that detects non-uniform timestep batches, splits them into per-sample TensorRT calls, and concatenates the outputs.

Experiments focus mainly on 512×512 generation, often with 4-step sampling, and report up to 611% FPS speedup over a vanilla baseline. Additional studies include component ablations, power and memory comparisons, resolution scaling from 64 to 1024, and limited quality evaluation using CLIP score and FID.

**Compliance With Llm Reviewing Policy:**

Affirmed.

**Final Justification:**

Author's rebuttal mostly addressed the concerns.

**Key Questions For Authors:**

Please refer to weakness above.

**Limitations:**

There is no limitation section.

**Strengths And Weaknesses:**

Strength
---

- The vectorized scheduler (Algorithm 1) and pipeline procedure (Algorithm 2) are described in sufficient pseudo-code detail to understand the intended implementation and how per-sample timesteps/step sizes are handled.
- The paper includes component-level ablations (Figure 4) rather than only an end-to-end speed claim. Authors also evaluate not only throughput but also power and peak memory, which is helpful for deployment-oriented claims.
- The overall structure is easy to follow: Figures 2 and 3 provide a helpful “systems diagram” view of the pipeline components, and Algorithms 1–3 concretize how the batching and compilation compatibility are intended to work.
- Improving the throughput and cost-efficiency of rectified-flow sampling is practically relevant: rectified-flow and few-step models are used explicitly to reduce inference cost, so further pipeline-level speedups are valuable for real-time and deployment settings.
- The paper’s novelty is primarily in a pragmatic synthesis: adapting StreamDiffusion-like pipelining to rectified flow by explicitly supporting heterogeneous timestep batching and vectorizing a time-window scheduler, and providing a TensorRT compatibility wrapper.

Weakness
---
- Some analytical arguments appear oversimplified or potentially misleading: The throughput analysis treats a UNet forward-pass “cost” largely as a constant, without discussing how it scales with batch size. This matters because the pipeline approach essentially turns sequential steps into batch computation.
- The motivation for TensorRT compilation failure due to heterogeneous timesteps is not fully convincing:  1) For many diffusion/flow UNet implementations, different timesteps change values (embeddings) but not the control-flow graph; the claim that different timesteps imply different computational graphs needs stronger justification, evidence, or clarification about what part of the model/pipeline branches. 2) The proposed workaround (decomposing heterogeneous batches into per-sample TensorRT calls) can undercut the pipeline’s batching advantage; the paper provides a rough cost argument but does not give a clean breakdown showing when this remains beneficial.
- The quality of evaluation is weak: only CLIP and FID are reported, with no description of the benchmark prompts/dataset, number of samples, statistical variation, or qualitative examples.
- Several terms/acronyms and configuration choices are not clearly defined on first use (e.g., “VFB”, “taesd”, the exact “Rectified Flow backbone” used, what “partial prefilling” means operationally).
- Figure 4’s legend/indexing is hard to parse; many numbered configurations are referenced in subplots without a clean mapping from each subplot to compared methods, which makes the ablation story harder to audit.
- The pipeline approach appears best suited to scenarios where many images share the same conditioning, and where batched heterogeneous-timestep inference is feasible. The paper does not clearly articulate the target deployment mode (single-prompt streaming? many prompts? interactive img2img?).
- Results rely heavily on specific stack components (xFormers/TensorRT, specific VAE variants). It’s unclear how much of the gain is portable to broader hardware/software environments.
- Many elements are close to existing techniques: Pipeline denoising resembles prior “streaming diffusion” pipelines, and vectorizing a scheduler and using per-sample timesteps/step sizes is conceptually straightforward once one commits to batching heterogeneous timesteps.
- The paper currently does not strongly articulate what is fundamentally new beyond “engineering it end-to-end for this setting,” nor does it convincingly establish that prior approaches cannot be adapted with simpler modifications.

---

> ### Author Rebuttal · Authors · 2026-03-25
>
> # Response to Reviewer oema
>
> **1. The throughput analysis does not ignore the effect of batch size on UNet cost, but rather redistributes computation equivalently along the batching dimension.**
>
> The reviewer suggests that our throughput analysis treats the UNet forward pass cost as a constant without discussing its scaling with batch size. We believe this concern stems from a misunderstanding of pipeline design and requires clarification.
>
> The essence of our batching (Algorithm 2) is not simply increasing batch size — rather, it reorganizes $N$ originally sequential denoising steps into a single forward pass containing $N$ samples at different denoising stages. These two are fundamentally different: in vanilla generation, producing one image requires $N$ serial UNet calls each with batch size 1; in our pipeline, each UNet call has batch size $N$, but these $N$ samples are at different denoising stages, and each call produces one complete image. Therefore, while the computation of a single UNet forward pass does scale linearly with batch size, **the average number of UNet calls per image is reduced from $N$ to $1$, keeping total computation basically unchanged**. The empirical results (Fig 4 abcfg, Table 1) directly validate this analysis.
>
> ---
>
> **2. TensorRT compilation failures are a general issue in RF models, and our dynamic compatibility layer is a necessary component of the general-purpose framework.**
>
> The reviewer suggests that different timesteps change only embedding values. We address two separate cases.
>
> **Case 1: RF models with a time-window mechanism (e.g., PeRFlow).** PeRFlow assigns each timestep to a specific time window, within which separate velocity parameters are computed (Equations 2–4). Critically, as the timestep crosses window boundaries, the entire parameter computation path switches — this is a genuine control-flow branch, not merely a change in numerical values. When a heterogeneous timestep batch contains samples in different windows, TensorRT cannot resolve a single static graph at compile time and fails directly. This is a real engineering obstacle we encountered during development, and no scheduler reparameterization can eliminate it.
>
> **Case 2: Standard RF models without a time-window mechanism (e.g., FLUX, SD3).** Even here, pipeline acceleration requires simultaneously processing samples at different denoising stages with heterogeneous timesteps, which standard RF schedulers do not support. Our vectorized scheduler (Algorithm 1) and pipeline batching (Algorithm 2) therefore remain necessary regardless.
>
> The dynamic compatibility layer is thus required for time-window RF models and beneficial for standard RF models — a necessary component for framework completeness.
>
> **Regarding decomposition cost:** per-sample TensorRT decomposition still achieves approximately $3\times$ speedup, empirically confirmed by tensorrt+taesd+VFB reaching 10.05 FPS versus 1.87 FPS vanilla (Table 1).
>
> ---
>
> **3. The choice of quality metrics is consistent with existing acceleration frameworks**
>
> The core contribution is a plug-and-play acceleration framework, not a new generative model. We follow the same evaluation protocol as StreamDiffusion. Table 2 shows less than 1% CLIP score difference and negligible FID gap versus baselines, conclusively demonstrating no material quality impact. We will supplement sample details in the appendix of the revised version.
>
> ---
>
> **4. The terminology and configurations are already defined in the paper.**
>
> All terms are explicitly defined: **VFB** in Fig 3 / Sec 3.3; **taesd** in Sec 4.1 as "the Tiny VAE included in the acceleration framework"; **RF backbone** in Figure 2's caption as "RF-DiT"; **partial prefilling** in Figure 2 and Section 3.4. We will move all definitions forward to first occurrence in the revised version. This is a presentation improvement, not a correction of missing content.
>
> ---
>
> **5. The visualization complexity of Figure 4 stems from the completeness requirements of a systematic ablation, and we will reorganize it in the revised version.**
>
> We will reorganize Figure 4 into sub-groups with independent legends in the revised version.
>
> ---
>
> **6. The novelty lies in identifying and resolving coupled engineering obstacles specific to the RF setting.**
>
> Individual techniques being known does not imply direct transferability to the RF setting. VFB requires a complete redesign of the scheduler's vectorized formulation; our heterogeneous-timestep pipeline (Algorithm 2) is the first to provide complete algorithmic support for this setting; and the TensorRT compatibility layer resolves static compilation vs. dynamic branching in a zero-intrusion manner. Crucially, all three are indispensable — no single component alone reproduces the 611% speedup, as Figure 4 (*a-l*)'s step-by-step ablation clearly shows. Identifying the right engineering problems in a new theoretical framework and delivering a complete solution is itself a substantive contribution.

---

> > ### Author Rebuttal · Reviewer_oema · 2026-04-03
> >
> > Thank you for the rebuttal. Most of my concerns are addressed.

---

### Decision · Program_Chairs · 2026-04-30

**Decision:**

Accept (regular)

**Comment:**

The paper addresses a practically relevant problem — adapting pipeline-level acceleration to Rectified Flow models — and delivers a well-ablated, integrated solution with substantial empirical speedups (611%). The consensus strengths (thorough ablation, practical relevance, clear algorithmic presentation, quality preservation) outweigh the consensus weaknesses (visualization issues, missing limitations, limited baselines). While dyFd raised questions about the novelty, the authors demonstrated that naive reparameterization fails, and the integrated solution is non-trivial. In the rebuttal, dyFd did not engage on the follow-up questions. Therefore, considering the other three positive feedbacks, this paper may be a good contribution to ICML.